# Next-Generation Sequencing of Local Romanian Tomato Varieties and Bioinformatics Analysis of the *Ve* Locus

**DOI:** 10.3390/ijms23179750

**Published:** 2022-08-28

**Authors:** Anca-Amalia Udriște, Mihaela Iordachescu, Roxana Ciceoi, Liliana Bădulescu

**Affiliations:** 1Research Center for Studies of Food Quality and Agricultural Products, University of Agronomic Sciences and Veterinary Medicine of Bucharest, 59, Mărăști Bd., 011464 Bucharest, Romania; 2Faculty of Horticulture, University of Agronomic Sciences and Veterinary Medicine of Bucharest, 59, Mărăști Bd., 011464 Bucharest, Romania

**Keywords:** *Solanum lycopersicum* L., NGS, genetic variability, biotic stress, Romanian tomato, *Verticillium* wilt

## Abstract

Genetic variability is extremely important, not only for the species’ adaptation to environmental challenges, but also for the creation of novel varieties through plant breeding. Tomato is an important vegetable crop, as well as a model species in numerous genomic studies. Its genome was fully sequenced in 2012 for the ‘Heinz 1706’ variety, and since then, resequencing efforts have revealed genetic variability data that can be used for multiple purposes, including triggering mechanisms of biotic and abiotic stress resistance. The present study focused on the analysis of the genome variation for eight Romanian local tomato varieties using next-generation sequencing technique, and as a case study, the sequence analysis of the *Ve1* and *Ve2* loci, to determine which genotypes might be good candidates for future breeding of tomato varieties resistant to *Verticillium* species. The analysis of the *Ve* locus identified several genotypes that could be donors of the *Ve1* gene conferring resistance to *Verticillium* race 1. Sequencing for the first time Romanian genotypes enriched the existing data on various world tomato genetic resources, but also opened the way for the molecular breeding in Romania. Plant breeders can use these data to create novel tomato varieties adapted to the ever-changing environment.

## 1. Introduction

Tomato (*Solanum lycopersicum* L.) fruits have substantial nutraceutical qualities, being an important source of fibers, lycopene and other carotenoids, vitamin C, and potassium, their consumption reducing the risk of certain cancers, cardiovascular disease, ultraviolet-light-induced skin damage, and osteoporosis [1,2]. Additionally, tomato crop production is important from an economical point of view, as it is the second most important fruit or vegetable crop next to potato (*Solanum tuberosum* L.) being cultivated worldwide [3].

Tomato belongs to the Solanaceae family, and it has been used as a model in molecular genetic and genomic studies for fruiting plants because of its diploid, compact genome (about 950 Mb) [4]. Thus, the tomato whole genome of the ‘Heinz 1706’ variety has been the first to be sequenced, after a 10-country and 8-year collaborative effort, using a combination of classical Sanger sequencing and the emerging new NGS sequencing, the work being completed in 2012 [5,6].

Whole-genome resequencing via Illumina platforms, based on the mechanism of SBS (sequencing by synthesis), has become the most rapid and effective method to identify the genetic variations in individuals of the same species or between related species. The various data, such as single nucleotide polymorphism (SNP), insertion and deletion (InDel), copy number variation (CNV), and structural variation (SV), obtained through resequencing is used in population genetics research and genome-wide association studies (GWAS) to investigate the mechanisms of biotic and abiotic stress resistance, to select plants and animals for agricultural breeding programs, to identify common genetic variations among populations, and more [7,8,9,10].

Whole-genome resequencing (WGRS) analysis represents a powerful strategy for rapid identification of candidate genes responsible for traits of interests [7,11,12,13]. One of the fungal diseases that affects many crops, causing yield and quality losses, is *Verticillium* wilt, due to infection with *Verticillium* sp. In tomato, two linked genes providing resistance to *Verticillium dahliae* race 1, *Ve1* and *Ve2*, located in the *Ve* locus in chromosome 9, have been cloned, genes putatively encoding cell surface-like receptors [14]. Fradin et al., in 2009, showed that, out of the two genes, *Ve1* was the one actually providing the resistance to *V. dahliae* and *V. albo-atrum*. Sequence analysis of the two genes revealed that one deletion in *Ve1* resulted in the production of a truncated protein, which was present in all susceptible genotypes analyzed, thus providing a putative marker that could be used by the plant breeders to discriminate between the resistant and susceptible genotypes [15].

In the present study, we analyzed genome variation from the perspective of number and distribution of SNPs, InDels, SVs, and CNVs for eight Romanian local tomato varieties. In addition, as a case study, we analyzed the sequence of *Ve1* and *Ve2* loci in these Romanian tomato genotypes in order to determine which of them might be good candidates for future breeding of tomato varieties resistant to *Verticillium* species.

## 2. Results

### 2.1. NGS Data Analysis

#### 2.1.1. Sequencing Data Quality Control

The genomes of eight Romanian tomato varieties were sequenced using NGS technology. Sequencing quality distribution was examined over the full length of all sequences, to detect any sites with an unusually low sequencing quality, where incorrect bases may have been incorporated at abnormally high levels. Q30 is considered a benchmark for quality in next-generation sequencing [16]. For the present sequencing, data results showed that Q30 was over 90% for all studied genotypes, and the ratio of clean data to raw data (effective rate) was around 99% (Appendix A).

#### 2.1.2. SNP Detection, Distribution, and Mutation Frequency

SNP variations were detected in all eight studied genotypes; however, their number and distribution within the genomes varied among the genotypes.

The SNPs were distributed in all regions of the genomes: upstream, exonic (stop gain, stop loss, synonymous, nonsynonymous), intronic, splicing, downstream, upstream/downstream, intergenic, others (Appendix A). A total of 2,964,636 SNPs were identified within the genotypes studied, ranging from 223,072 in Buzău 1600 to 697,473 in Florina 44. The highest numbers of SNPs were detected in the intergenic regions for all genotypes (between 190,064/85.14% for Buzău 1600 and 623,805/89.42% for Florina 44) (Appendix A).

For all studied genotypes, the number of transitions (ts), point mutations that change a purine nucleotide to another purine or a pyrimidine nucleotide to another pyrimidine, was higher than the number of transversions (tv), point mutations that substitute a purine for a pyrimidine or vice versa. However, the ratio ts/tv was similar for all genotypes (between 1.294 for Florina 44 and 1.394 for Ștefănești 24). For the exonic SNPs, for all genotypes studied, the number of nonsynonymous SNPs was higher than the synonymous ones, the lowest number of nonsynonymous SNPs being observed for Buzău 1600 (3010/1943, respectively, 58.42%/37.71%) and the highest for Florina 44 (5904/4062 respectively 57.51%/39.57%). Stop gain point mutations were also more numerous than stop loss ones in all genotypes under study, again the least numerous being observed in Buzău 1600 (78/23, respectively, 1,51%/0.45%), and the highest number being observed in Florina 44 (153/46, respectively, 1.49%/0.45%). The highest heterozygosity rate (‰) was observed for the genotype Kristinica (0.234), whereas the lowest rate was observed for the genotype Buzău 47 (0.115).

The distribution of the six types of SNP mutations is displayed in Figure 1. The highest number of SNPs was observed in the genotype Florina 44 for all six SNP types, and the lowest number was observed for the genotype Buzău 1600. Among the six types of SNP mutations, for all genotypes, the highest number of SNPs was the T:A > C:G type, followed by C:G > T:A, whereas the lowest number of SNPs was the C:G > G:C type.

#### 2.1.3. Insertion/Deletion Detection and Distribution

InDel variations were detected in all eight studied genotypes, and like in the case of SNPs, their number and distribution within the genomes varied from genotype to genotype.

The InDels were distributed in all regions of the genomes: upstream, exonic (stop gain, stop loss, synonymous, nonsynonymous), intronic, splicing, downstream, up-stream/downstream, intergenic, others (Appendix A). A visual representation of these data is visible in Appendix A.

A total of 622,988 InDels were identified within the studied genotypes, ranging from 66,768 in Buzău 1600 to 110,138 in Florina 44. For all genotypes, the total number of insertions was roughly double the total number of deletions. The highest number of InDels was detected in the intergenic regions for all genotypes (between 47,309 in Buzău 1600 and 82,716 in Florina 44), followed by the intronic (between 8360 in Buzău 1600 and 12,236 in Florina 44), upstream (between 4584 in Argeș 20 and 6341 in Florina 44), downstream (between 2915 in Buzău 47 and 4157 in Florina 44), upstream/downstream (between 357 in Buzău 47 and 578 in Florina 44), exonic (between 338 in Buzău 1600 and Buzău 47 and 493 in Florina 44), and splicing regions (between 32 in Buzău 47 and 52 in Florina 44). For the exonic InDels, for all genotypes, the number of frameshift InDels were higher than the non-frameshift ones. The highest number of frameshift deletions was detected in the Florina 44 genotype (163), and the lowest number in the genotype Buzău 1600 (115), and for the frameshift insertions the highest number was observed in genotype Ștefănești 24 (129) and the lowest number in the genotype Buzău 47 (102). As for the non-frameshift InDels, the highest number of deletions was detected in the genotype Ștefănești 24 (101) and the lowest number in the genotype Buzău 47 (47).

InDels distribution within the coding sequence is portrayed in Figure 2. The highest number of InDels was observed for the 1 bp insertion/deletion and decreased with the increase in sequence length. Thereafter, the percentages for sequences with lengths multiples of three bp were higher than those of other lengths. The reason for these higher percentages might be that these sequences do not cause frameshifts and, subsequently, premature STOP codons. InDels with lengths beyond 19 bp were below 1%.

InDels distribution within the whole genome is presented in Figure 3. The highest percentage of InDels was again observed for the 1 bp insertion/deletion and decreased gradually with the increase in sequence length.

For the eight studied genotypes, the SNP and InDel densities on each chromosome were similar (Figure 4) and varied as follows. For the genotype Kristinica, the highest density was noted in chromosome 11, followed by chromosome 4. InDel density was also noted in chromosomes 2, 5, and 9.

For the Florina 44 variety, the highest density was observed in chromosome 4, followed by chromosome 11. For the Andrada variety, the highest density was observed in chromosome 4, but high densities were also noted for chromosomes 2, 5, 9, and 11. For the Buzău 1600 variety, the highest density was observed in chromosome 4, with high density being present as well in chromosomes 5, 9, and 11. In the case of the Buzău 47 variety, chromosome 11 presented the highest density, and high density existing as well in chromosomes 4, 5, 8, and 9. For the variety Argeș 11, the highest density was observed in chromosome 11, but high densities were also observed in chromosomes 2, 3, 4, 5, 9, and 10. For the Argeș 20 variety, again the highest density was noted in chromosome 11, with high densities in chromosomes 2, 4, 5, 7, 9, and 10. Lastly, Ștefănești 24 presented high densities in chromosomes 4, 11, and 12, and, to a lesser extent, in chromosomes 1, 5, 6, and 7.

#### 2.1.4. Structural Variant Detection and Annotation

Structural variations were detected in all eight studied genotypes, and their number and distribution within the genomes varied from genotype to genotype.

The SVs were distributed in all regions of the genomes: upstream, exonic, downstream, intronic, upstream/downstream, splicing, intergenic, and others (Appendix A). A visual representation of these data is visible in Appendix A. The highest number of SVs was observed in the intergenic regions (between 1623 in Argeș 20 and 2663 in Andrada), followed by the exonic (between 446 in Argeș 20 and 817 in Florina 44), intronic (between 96 in Buzău 47 and 169 in Andrada), upstream (between 55 in Buzău 47 and 109 in Andrada), downstream (between 32 in Argeș 20 and 74 in Andrada), upstream/downstream (between 6 in Buzău 1600 and 11 in Florina 44), and splicing (between none in Buzău 1600 and Argeș 11 and 3 in Argeș 20) regions. The distribution of the five types of SVs is visible in Figure 5.

The highest percentage of SVs was interchromosomal translocations, followed by the deletions, intrachromosomal translocations, inversions, and insertions. More than 40% of SVs were longer than 1200 bp. Approximately 20% of the SVs had a length of 200–300 bp. For the rest of the SVs’ length-size categories, the percentages were lower than 7%. The rarest SVs were those with a length of less than 100 bp (~0.4–0.7%) (Figure 6).

#### 2.1.5. Copy Number Variations Detection and Annotation

For all genotypes studied, the number of deletions (between 7247-Argeș 11 and 13,350-Andrada) was higher than the number of duplications (between 1265-Buzău 1600 and 1824-Argeș 20). The highest numbers of CNVs were detected within the intergenic regions, followed by the exonic regions. The lowest numbers of CNVs were noted within the upstream/downstream regions (Appendix A). A visual representation of these data is visible in Appendix A.

### 2.2. Sequence Analyses of Ve1 and Ve2 Loci in Romanian Tomato Genotypes

To identify the sequences of *Ve1* and *Ve2* homologous genes in the studied genotypes, the NGS bam files containing the eight genomes were aligned with the reference genome using the Workbench software. For the *Ve1* locus, the genotypes Kristinica, Florina 44, Buzău 1600, Argeș 11, and Argeș 20 were identical with the gene from the Heinz 1706 reference genome. For the *Ve2* locus, the only genotypes that showed differences compared with the reference genome were Andrada and Buzău 47. Thereafter, the nucleotide sequences for the eight genotypes were aligned with previously published sequences of the genes.

For *Ve1*, the eight Romanian genotypes were aligned and compared with the following sequences: NC_015446.3, reference genome sequence of Heinz 1706 [17]; AF272366.2, of the Ailsa Craig genotype [14]; FJ464557.1, of the VFN-8 genotype; FJ464556.1, of the Motelle genotype; FJ464555.1, of the Moneymaker genotype; FJ464554.1, of the Craigella GCR26 genotype; and FJ464553.1, of the Craigella GCR218 genotype [15]. In the case of the *Ve1* locus, 9 SNPs were identified (Appendix A). In the case of the first 2 SNPs, only the genotype Ailsa Craig was different, with a cytosine inserted at position 29 and also a cytosine deleted at position 35, resulting in a PMV translation instead of LWL. At position 246, the SNP presents a silent mutation, G/C. At position 380, the SNP C/A resulted in an A/D amino acid translation. The genotypes Andrada, Buzău 47, Ailsa Craig, Moneymaker, and Craigella GCR25 had a cytosine at this position, whereas the rest contained an A. At position 610, the SNP A/T translated into a T/S amino acid. Ailsa Craig, Moneymaker, and Craigella GCR26 had an adenine at this position, while the rest contained a thymine. At position 706, there is another SNP A/T, again translated into T/S; however, this time, A is present in Andrada, Buzău 47, Ștefănești 24, Ailsa Craig, Moneymaker, and Craigella GCR26. A single nucleotide deletion exists at position 1220, resulting in a premature stop codon in Andrada, Buzău 47, Ștefănești 24, Moneymaker, and Craigella GCR26. At position 1548, the SNP C/G translates into N for the varieties that do not have a deletion at position 1220. All the varieties that contain guanine at this position are producing the truncated Ve1 protein. Lastly, at position 1888, the SNP G/A translates only into D, since all the varieties that contain adenine at this position are producing the truncated Ve1 protein due to the deletion at position 1220 (Table 1).

In the case of *Ve2*, the eight Romanian genotypes were aligned and compared with the following sequences: NC_015446.3, of the reference genome sequence Heinz 1706 [17]; AF365930.1, of the Ailsa Craig genotype [14]; FJ464562.1, of the genotype VFN-8; FJ464561.1, of the Motelle genotype; FJ464560.1, of the Moneymaker genotype; FJ464559.1, of the Craigella GCR218 genotype; and FJ464558.1 of the Craigella GCR26 genotype [15]. Again, 9 SNPs were identified (Appendix A). The first SNP, G/C, at position 1385, is translated into an R/T amino acid, the Romanian genotypes Andrada and Buzău 47 being the only ones that contain threonine. The second, at position 1811, C/T, translates into A/V, with Andrada, Buzău 47, Moneymaker, Craigella GCR26, and Ailsa Craig containing thymine. At position 2761, the SNP G/A translates into D/N, with Moneymaker being the only one that has adenine. At position 2771, the SNP C/G translates into T/R, with Andrada, Buzău 47, Moneymaker, Craigella GCR26, and Ailsa Craig having guanine. At position 2893, the SNP C/T translates into P/S, with Andrada, Buzău 47, VFN-8, Motelle, and Craigella GCR218 containing cytosine. The next two SNPs, at positions 2934 and 3243, are silent. Finally, at positions 3380 and 3383, T/C translates into F/S, with only Ailsa Craig containing thymines, TTTTTT vs. TCTTCT in the other genotypes (Table 2).

## 3. Discussion

The first tomato genome to be sequenced, Heinz 1706, provided the ‘golden standard’ for future resequencing efforts [6]. With the advent of NGS, more and more tomato genotypes have been wholly sequenced, enriching the knowledge at the DNA level and offering new data that can be used in subsequent studies, as well as in breeding for improved tomato varieties. The present study contributes to this growing pool of sequenced genomes with whole-genome resequencing data from eight Romanian tomato genotypes.

SNP data mined from sequenced transcriptomes and from resequenced whole genomes through next-generation sequencing have been used to study the diversity within cultivated tomato genotypes, as well as between cultivated tomatoes and wild-type relatives [12,18,19,20,21,22]. The present study reports almost 3 million SNPs, adding to/confirming those reported by the 100 Tomato Genome Sequencing Consortium [12] and Causse et al., 2013.

SNPs and InDels were not evenly distributed within the genome, for each genotype existing certain ‘hot spots’, where there was a higher density of SNPs/InDels, mostly toward the ends of chromosomes, but there were also observed wide regions with a high density of SNPs/InDels spanning almost the whole chromosome: chromosome 11 in all genotypes except Andrada, Buzău 1600, and Ștefănești 24; chromosome 4 in Florina 44; and chromosome 6 in Ștefănești 24. Interestingly, for each genotype, the ‘hot spots’ for SNPs and InDels overlapped (Figure 4). The higher polymorphism towards the chromosomes’ ends can be explained by the higher recombination frequency of these regions [19,23]. The broad regions with high polymorphism density were also observed in other genotypes in previously published studies, but on different chromosomes [20], most probably due to the introgressions from wild-type relatives, depending on each genotype breeding history. For instance, chromosome 11 of the Heinz 1706 genotype contains large introgressions from *S. pimpinellifolium*, having received them through disease resistance [6].

The highest numbers of SNP types were associated with T:A > C:G and C:G > T:A transitions. The prevalence of transitions as opposed to transversions has been observed in numerous other species, and is explained by the high frequency of the cytosine-to-thymine mutation following the deamination of methylated cytosine residues [24].

Structural variations, such as deletions, insertions, copy number variations, inversions, and translocations, play a major role in heritable phenotypic diversity within and between species, as they could lead to gene loss, gene duplication, and the creation of new genes [25]. If high-throughput short read sequencing is extremely efficient in detecting SNPs and small InDels, the short read length makes it difficult to characterize repetitive regions, and hence to detect efficiently structural variations [25]. Nevertheless, there are sequencing techniques that overcome these difficulties. For instance, long read nanopore sequencing significantly improves the success in identifying structural variations. If, in the present study, between ~7500 and 10,400 SV per genotype were identified, Alonge et al., 2020, identified almost 240,000 SVs in 100 tomato accessions. In addition, if in the present study, the highest numbers of SVs were translocations, in the above-mentioned study, the most common SVs were insertions and deletions.

Copy number variations are part of the structural variations. Most CNVs studied so far were those that affect protein-coding sequences, and thus result in either gains or losses of gene copies, and ultimately in the regulation of plant development and plant adaptation to environmental factors [26]. In the current study, the exonic detected CNVs were between 507 for Andrada and 673 for Argeș 20, with an average of 577, a value similar to that observed in the Causse et al. study, 2013 [20]. However, as mentioned before, with the complexity of the plant genomes added to the short read sequencing, their complete detection is difficult [26].

### Sequence Analyses of the Ve1 and Ve2 Loci in the Romanian Tomato Genotypes

The *Ve* locus in tomato comprises two genes that encode proteins involved in both stress/defense and plant growth [27]. *Ve1* expression is induced by various stress conditions, both biotic and abiotic, whereas *Ve2* is constitutively expressed [27].

In the case of the *Ve1* gene, the genotypes Andrada, Buzău 47, and Ștefănești 24 present the single nucleotide deletion at position 1220 that results in the premature stop codon and putative production of truncated protein. The presence of the *Ve1* allele in these genotypes implies that they are susceptible to *Verticillium* race 1. The other genotypes are identical at the amino acid level with Motelle, VFN-8, and Craigella GCR218 genotypes, which were proved to be resistant to *Verticillium* race 1, and thus good donors of the *Ve1* allele in future breeding programs.

In the case of the *Ve2* gene, the putatively resistant genotypes (Kristinica, Florina 44, Buzău 1600, Argeș 11, and Argeș 20) are identical at the amino acid level with the resistant genotypes Motelle, VFN-8, and Craigella GCR218, except for position 965, which contains a serine instead of a proline. If, initially, it was thought that only *Ve1* had a role in plant resistance to *Verticillium*, later, it was proved that the mechanism of resistance was more complex than originally thought, and both *Ve1* and *Ve2* are involved in the process. A study where *Ve2* gene expression was suppressed via RNAi demonstrated pronounced effects on defense/stress gene expression [28]. Even though the silencing of *Ve2* does not increase the susceptibility of either resistant or susceptible genotypes to *Verticillium*, in the resistant genotypes infected with *Verticillium* race 1, the silencing induces repression of multiple genes with a role on defense/stress, whereas in the susceptible genotypes that are missing a functional Ve1 protein, continuous Ve2 signaling is sufficient to produce a normal defense/stress response [28]. It remains to be seen in future studies if the change to serine at position 965 has a significant effect on the way the plants are coping with the *Verticillium* attack.

## 4. Materials and Methods

### 4.1. Plant Material

Eight Romanian tomato varieties, Kristinica, Florina 44, Andrada, Buzău 1600, Buzău 47, Argeș 11, Argeș 20, and Ștefănești 24, were analyzed in the present study. Tomato seeds received from the Vegetable Research and Development Station Buzău and the National Research and Development Institute for Biotechnology in Horticulture Ștefănești-Argeș were cultivated under greenhouse conditions (18–25 °C) in the Research Center for Studies of Food Quality and Agricultural Products of the University of Agronomic Sciences and Veterinary Medicine of Bucharest, Romania.

### 4.2. DNA Extraction

Genomic DNA was extracted from fresh leaves of tomato seedlings using an automated extraction system (InnuPure C16, Analytik Jena GmbH, Jena, Germany) based on the principle of magnetic particle separation for fully automated DNA isolation and purification. An InnuPREP Plant DNA I Kit-IPC16 (Analytik Jena GmbH, Jena, Germany) was used for genomic DNA extraction following the manufacturer’s instructions. A preliminary processing step was the external lysis of the starting material. The plant sample was ground to powder in the presence of liquid nitrogen and homogenized with SLS lysis solution (containing CTAB as detergent component), proteinase K, and RNase A solution. After external lysis, the extraction proceeded with automated DNA extraction following the manufacturer’s instructions. The DNA was quantified using a NanoDrop^TM^ 1000 spectrophotometer (Thermo Fisher Scientific, Wilmington, DE).

### 4.3. Sequencing, Computational Data Processing, and Sequencing Analysis

Whole-genome sequencing (WGS) was performed via an Illumina platform (NGS) by Novogene Co., Ltd., Cambridge, UK. An original image data file from the high-throughput sequencing platform Illumina was transformed to sequenced reads (raw data) by CASAVA base recognition (Base Calling) (Novogene Co., Ltd., Cambridge, UK). Raw data were stored in FASTQ (.fq) format files [29], which contain sequencing reads and corresponding base quality. The effective sequencing data were aligned with the reference sequence through the BWA (Li H. et al. 2009) software [30] (parameters: mem -t 4 -k 32 -M), and the mapping rate and coverage were counted according to the alignment results. In order to obtain clean reads, low-quality reads or reads with adaptors that would affect the quality of downstream analysis were removed (Novogene Co., Ltd., Cambridge, UK). The Phred score (Q_phred_), the quality score of a base, was calculated using the equation Q_phred_ = −10log_10_€, where ‘e’ represents the sequencing error rate.

The filtered reads were mapped onto the tomato genome SL3.0 [17], used as a reference sequence. The resultant sequence alignment format files were converted to binary sequence alignment format (*.bam) files and subjected to yield a variant file including SNP information. The mapping rates of samples reflect the similarity between each sample and the reference genome. The depth and coverage are indicators of the evenness and homology with the reference genome (Novogene Co., Ltd., Cambridge, UK).

#### 4.3.1. SNP Detection and Annotation

Individual SNP variations were detected using SAMtools with the ‘mpileup -m 2 -F 0.002 -d 1000’ parameter [31] (Novogene Co., Ltd., Cambridge, UK). To reduce the error rate in SNP detection, the results were filtered using two criteria: the number of support reads for each SNP was higher than 4, and the mapping quality of each SNP, calculated by the root mean square of the support reads’ mapping quality, was higher than 20. Thereafter, the SNPs were annotated using the ANNOVAR software [32] (Novogene Co., Ltd., Cambridge, UK) in the following categories: upstream (located within 1 kb upstream away from transcription start site of the gene), exonic (located in the exonic region), intronic (located in the intronic region), splicing (located in the splicing site, within a 2 bp range of the intron/exon boundary), downstream (located within 1 kb downstream away from transcription termination site of the gene region), upstream/downstream (located within the less than 2 kb intergenic region, which is in 1 kb downstream or upstream of the genes), intergenic (located within the more than 2 kb intergenic region), and others (located in other region). The exonic category was further split into nonsynonymous (single-nucleotide mutation with changing the amino acid sequence), synonymous (single-nucleotide mutation without changing the amino acid sequence), stop gain (a nonsynonymous SNP that leads to the introduction of a stop codon at the variant site), and stop loss (a nonsynonymous SNP that leads to the removal of the stop codon at the variant site). The genome-wide heterozygous rate for SNPs (het rate (‰)) was calculated as the ratio of heterozygous SNPs to the total number of genome bases.

Based on the type of mutations, the SNPs were classified into six categories: T:A > C:G, T:A > G:C, C:G > T:A, C:G > A:T, T:A > A:T, and C:G > G:C. For instance, the T:A > C:G mutations include mutations from T to C and A to G. When a T-to-C (T > C) mutation appears on either of the double strand, the A-to-G (A > G) mutation will be found in the same position of the other chain. Therefore, the T > C and A > G mutations were classified into a single category.

#### 4.3.2. Insertion/Deletion (InDel) Detection and Annotation

An InDel was defined as the insertion or deletion of a DNA sequence with a length of 50 bp or less. InDels were detected using SAMTOOLS [31] with the ‘mpileup -m 2 -F 0.002 -d 1000’ parameter and annotated using the ANNOVAR software [32] (Novogene Co., Ltd., Cambridge, UK). The filter conditions to reduce the error rate in InDel detections were the same as with the SNP detection.

The annotation of InDels was performed using the same categories for genomic regions as the SNPs, except for the exonic region, which was subdivided into stop gain and stop loss (same as SNPs), frameshift deletion and frameshift insertion (InDel mutation changing the open reading frame with deletion or insertion), and non-frameshift deletion and non-frameshift insertion (InDel mutation without changing the open reading frame with deletion or insertion sequences of 3 or multiple of 3 bases).

Length distribution of InDels was analyzed as a percentage within the coding sequence (CDS) and within the whole genome.

#### 4.3.3. Structural Variant Detection and Annotation

Structural variants (SVs) were defined as genomic variations with mutations of a relatively larger size, more than 50 bp, such as deletions (DEL), insertions (INS), inversions (INV), intrachromosomal translocations (ITX), and interchromosomal translocations (CTX) and were detected by the BreakDancer software [33]. SVs that were not supported by at least two pair-end read alignments were removed from further analysis. The insertions, deletions, and inversions were annotated by the ANNOVAR software [32].

#### 4.3.4. Copy Number Variation Detection and Annotation

Copy number variations (CNV) were defined as structural variations showing deletions or duplications in the genome. Based on the reads’ depth of the reference genome, the CNVnator software [34] was used to detect CNVs of potential deletions and duplications with the parameter ‘-call 100’. The detected CNVs were further annotated by the ANNOVAR software [32].

### 4.4. Sequence Analysis of the Ve Locus

Next-generation sequencing BAM files containing the nucleotide sequence data for the eight tomato varieties studied were loaded onto the Workbench software and aligned to the reference genome. For each variety, the differences in nucleotide sequence were noted. Amino acid sequences for each variety were generated using the Sequence Manipulation Suite: ORF Finder software [35].

Nucleotide sequences of *Ve* genes and amino acid sequences of corresponding putative proteins from the Romanian genotypes included in this study and sequences of *Ve* genes reported previously were aligned using the MultAlin software [36].

## 5. Conclusions

In the present times, we face a race between plant breeders on the one hand, who are creating new crop plant varieties that are resistant or at least tolerant to pathogens and viruses, and biotic factors on the other hand, which are constantly mutating and developing new races/strains that overcome plant resistance [37]. The resequencing of new *L. esculentum* varieties will enable researchers to link phenotypical variations to their DNA sequence variation, uncovering new information for comparative genomics studies [38]. Therefore, rather than being an end point, by bringing up novel essential data, NGS brings to light a plethora of new questions and opens up new research directions. Some of the varieties studied, such as Buzău 1600 and Buzău 47, were created between 1970 and 1980, and are still appreciated by growers and consumers alike, possessing multiple traits that would recommend them as genitors in tomato breeding [39]. One of the goals of the research founded by the Romanian Ministry of Agriculture and Rural Development, of which this study is a part of, is to create a database of Romanian cultivars/varieties’ genetic variations that could be used in the future by plant breeders for selecting genitors that could donate genes encoding desirable traits. As an *in silico* case study, the survey of the *Ve* locus permitted the selection of a number of genotypes (Kristinica, Florina 44, Buzău 1600, Argeș 11, and Argeș 20) that could be donors of the *Ve1* gene conferring resistance to *Verticillium* race 1 attack, since they have amino acid sequences identical to those of proven resistant genotypes. The selected genotypes will be assessed for the confirmation of fungal resistance by artificial inoculation with different races of *Verticillium* prior to their use in plant breeding.

## Figures and Tables

**Figure 1 ijms-23-09750-f001:**
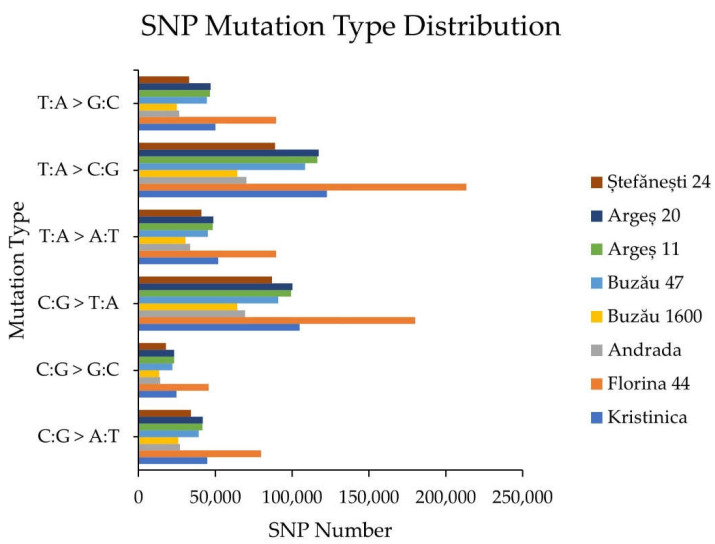
SNP mutation type distribution.

**Figure 2 ijms-23-09750-f002:**
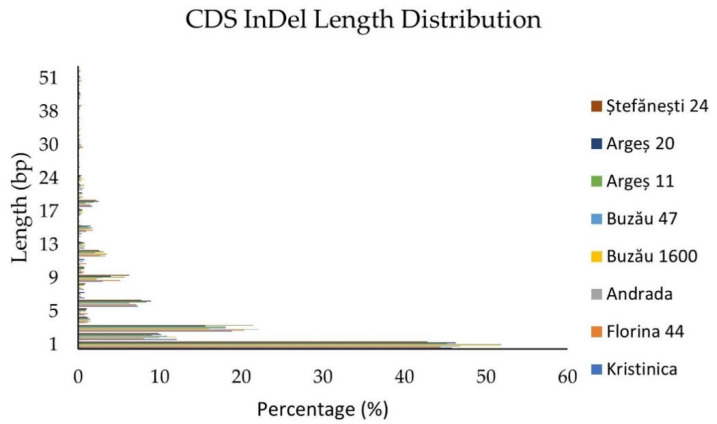
InDels distribution within the coding sequence.

**Figure 3 ijms-23-09750-f003:**
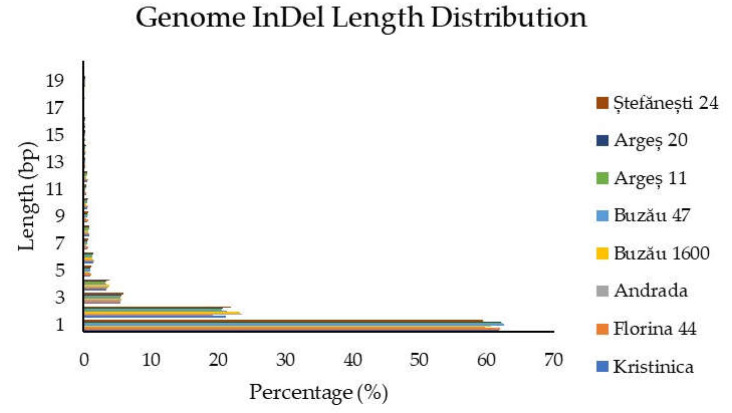
InDels distribution within the genome.

**Figure 4 ijms-23-09750-f004:**
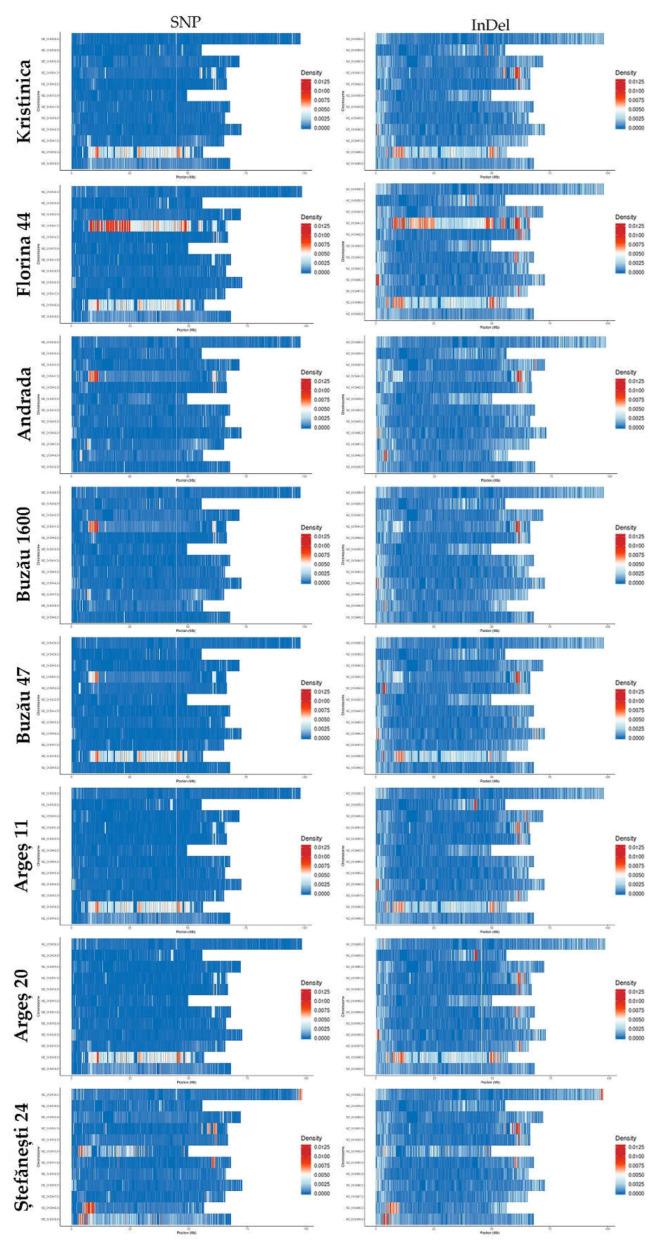
SNP and InDel densities per chromosome per genotype.

**Figure 5 ijms-23-09750-f005:**
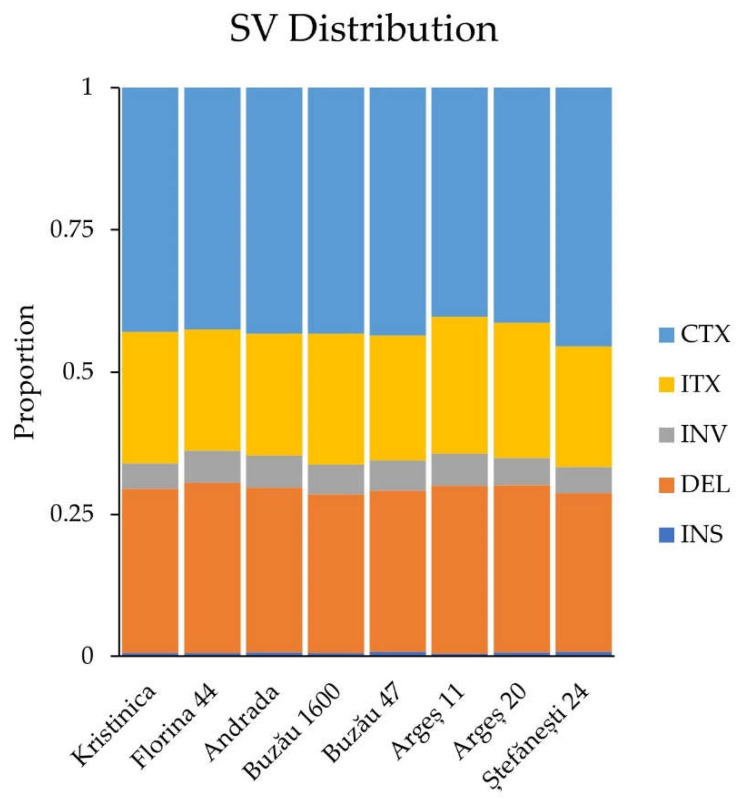
SV distribution within genotypes studied. CTX—interchromosomal translocation, DEL—deletion, INS—insertion, INV—inversion, ITX—intrachromosomal translocation.

**Figure 6 ijms-23-09750-f006:**
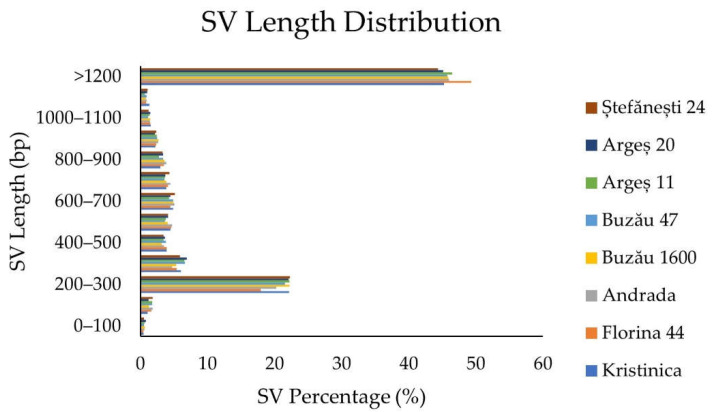
Structural variations’ (SVs) length distribution.

**Table 1 ijms-23-09750-t001:** Sequence analysis of the *Ve1* gene.

SNP Position	DNA Sequence	Amino Acid Sequence	Genotypes									
			Heinz 1706	Craigella GCR218	Motelle	VFN-8	Ailsa Craig	Craigella GCR26	Moneymaker	Kristinica, Florina 44, Buzău 1600, Argeș 11, Argeș 20	Andrada, Buzău 47	Ștefănești 24
29/35	CCTATGGTT	PMV		-	-	-	+	-	-	-	-	-
	CTATGGCTT	LWL	+	+	+	+	-	+	+	+	+	+
246	GTG	Silent	+	+	+	+	-	-	-	+	-	+
	GTC		-	-	-	-	+	+	+	-	+	-
380	GAC	D	-	-	+	-	+	+	-	-	+	-
	GCC	A	+	+	-	+	-	-	+	+	-	+
610	ACT	T	+	+	+	+	-	-	-	+	+	+
	TCT	S	-	-	-	-	+	+	+	-	-	-
706	ACT	T	-	-	-	-	+	+	+	-	+	+
	TCT	S	+	+	+	+	-	-	-	+	-	-
1220	TCAGAG	SE	+	+	+	+	-	-	-	+	-	-
	**TAG**AG	STOP	-	-	-	-	+	+	+	-	+	+
1548	AAC	N	+	+	+	+	-	-	-	+	-	+
	AAG	K *	-	-	-	-	+	+	+	-	+	-
1888	GAC	D	+	+	+	+	-	-	-	+	-	-
	AAC	N *	-	-	-	-	+	+	+	-	+	+

* The amino acids are not translated due to the STOP codon positioned upstream of these sequences. “-“/“+” denotes the absence/presence of the SNP. The underlined sequence encodes the STOP codon.

**Table 2 ijms-23-09750-t002:** Sequence analysis of the *Ve2* gene.

SNP Position	DNA Sequence	Amino Acid Sequence	Genotypes									
			Heinz 1706	Craigella GCR218	Motelle	VFN-8	Ailsa Craig	Craigella GCR26	Moneymaker	Kristinica, Florina 44, Buzău 1600, Argeș 11, Argeș 20	Andrada, Buzău 47	Ștefănești 24
1385	ACA	T	-	-	-	-	-	-	-	-	+	-
	AGA	R	+	+	+	+	+	+	+	+	-	+
1811	GTA	V	-	-	-	-	+	+	+	-	+	
	GCA	A	+	+	+	+	-	-	-	+	-	+
2761	GAC	D	+	+	+	+	+	+	-	+	+	+
	AAC	N	-	-	-	-	-	-	+	-	-	-
2771	AGA	R	-	-	-	-	-	+	+	-	+	-
	ACA	T	+	+	+	+	+	-	-	+	-	+
2893	CCA	P	-	+	+	+	-	-	-	-	+	-
	TCA	S	+	-	-	-	+	+	+	+	-	+
2934	CTC	Silent	-	-	-	-	+	+	+	-	-	-
	CTT		+	+	+	+	-	-	-	+	+	+
3243	GGT	Silent	-	-	-	-	+	+	+	-	+	-
	GGG		+	+	+	+	-	-	-	+	-	+
3380	TTT	T	-	-	-	-	+	-	-	-	-	-
	TCT	S	+	+	+	+	-	+	+	+	+	+
3383	TTT	T	-	-	-	-	+	-	-	-	-	-
	TCT	S	+	+	+	+	-	+	+	+	+	+

“-“/“+” denotes the absence/presence of the SNP.

## Data Availability

Not applicable.

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
