# Peer review of "Next-Generation Sequencing of Local Romanian Tomato Varieties and Bioinformatics Analysis of the Ve Locus"

_ijms, 2022, doi:10.3390/ijms23179750_

Round 1

Reviewer 1 Report

The manuscript submitted for review is written correctly and clearly. There are no downsides to my knowledge. However, I am not a genomics specialist, but a phytopathologist. From my point of view, it would be interesting to artificially inoculate the pathogen and verify the resistance to this pathogen of selected individuals with a particular SNP

Author Response

Dear Reviewer,

We thank you for taking the time to revise our manuscript. Your comment regarding the artificial inoculation of the putative resistant plants is more than appreciated, as we were also considering doing these tests in a follow-up project, that we hope it will be founded, as we stated in a phrase added to the conclusions of the manuscript. 

Reviewer 2 Report

This study analyzed eight Romanian local tomato varieties for distribution of SNPs, InDels, SVs, and CNVs. In addition, the authors analyzed the sequence of Ve1 and Ve2 loci in these Romanian tomato genotypes to determine which might be good candidates for future breeding of tomato varieties resistant to Verticillium species. Even though NGS is getting cheaper and cheaper, there is a need to address why whole genome sequencing was conducted instead of PCR to amplify only the V1 and V2 regions, sequence them, and analyze them for SNP marker development and marker-assisted breeding. 

Author Response

Dear Reviewer,

We thank you for taking the time to revise our manuscript. In answer to your comment regarding the reason for doing NGS instead of PCR of the desired region followed by Sanger sequencing, is that this study is part of a bigger project founded by the Romanian Ministry of Agriculture  and Rural development, and one of the goals of this project is to create a database of Romanian cultivars/varieties genetic variations that could be used in the future by plant breeders, for selecting genitors that could donate genes encoding desirable traits. So, we focused in presenting the results of NGS required by the project and in addition, we present the analysis of Ve locus as a case study, and we stated as such more clearly in the conclusions of the manuscript.

Reviewer 3 Report

Dear Authors

The manuscript includes a very impressive study that showed a variation among tomato varieties at genomic level

Author Response

Dear Reviewer,

We thank you for taking the time to revise our manuscript and accepting it as such. We are glad you find it interesting, and you appreciated our work.

Reviewer 4 Report

 I accept without any modifications. 

Author Response

(The authors gave the same response as above.)
